# Characterization of Root Morphology and Anatomical Structure of Spring Maize under Varying N Application Rates and Their Effects on Yield

Xiangling Li [1,2], Rui Wang [1,2], Baoyuan Zhou [1], Xinbing Wang [1], Jian Wang [2], Ming Zhao [1] and Congfeng Li [1,*]

[1] Institute of Crop Sciences, Chinese Academy of Agricultural Sciences/Key Laboratory of Crop Physiology and Production, Ministry of Agriculture and Rural Affairs, Beijing 100081, China
[2] College of Agronomy and Biotechnology, Hebei Normal University of Science & Technology/Hebei Key Laboratory of Crop Stress Biology, Qinhuangdao 066000, China
* Correspondence: licongfeng@caas.cn

**Abstract:** Root morphology is an important factor determining nitrogen (N) uptake by plants, which might be affected by the extent of N application. The processes associated with root morphogenesis of spring maize in response to N application rates remain poorly understood. In this study, both field and pot experiments were conducted to explore the effect of zero-N (N0), optimized-N (N180), and high-N (N360) on root morphology, anatomical structure, and N accumulation in spring maize. N application rates affected root length and surface area, and its endogenous hormone contents. The largest difference in total root length and surface area among the three N rates was found at the silking stage: the total root length and surface increased by 51.36% and 42.58% under N180 and by 7.8% and 30.14% under N360, respectively, compared with N0, and the root/shoot ratio and root bleeding sap significantly increased under N180 and N360 compared with N0. The auxin and jasmonic acid levels of roots under N180 and N360 were higher than N0. N application rates also affected root microstructure and ultrastructure. Compared with N0, the proportions of root aerating tissue under N180 and N360 were decreased by 32.42% and 11.92% at silking. The root tip cell structure was damaged under N0, and intact under N180 and N360. Moreover, the $^{15}N$ allocation proportions to root and grain under N180 and N360 were increased compared to N0. Grain yields under N180 and N360 increased by 20.44% and 16.6% compared with N0, respectively. It can be concluded that optimized-N application decreased root aerated tissue and thus improved root length and root surface area through regulating auxin and jasmonic acid levels and affected N uptake and grain yield of N-efficient spring maize variety.

**Keywords:** spring maize; root morphology; root anatomical structure; nitrogen uptake efficiency; grain yield

## 1. Introduction

Currently, many people throughout the world face food insecurity, especially since the COVID-19 pandemic has slowed down human activities globally and adversely affected global food production and aggravated the threat to food security [1,2]. Maize, one of the staple crops in the world, has great total production and more grain yields, and it thus holds an important role in ensuring food security [3].

Nitrogen (N) is the most important nutrient element for crop growth. The over-use of chemical fertilizers has played an important role to increase crop yield [4,5]. However, excessive application of fertilizer has brought about unstable corn yield, reducing N use efficiency, and environmental degradation [6–8]. Improving N use efficiency and reducing the input of N fertilizer has become a matter of great concern to scientists all over the world. N-efficient varieties are beneficial to improve grain yield and N use efficiency and reduce farmland environmental pollution [9,10].

The selection of high-yield crop varieties based on the root system was the second "green revolution" to ensure food security [11], which showed the importance of the root system to crop growth. Roots are critically important for the acquisition of water and mineral nutrients. Therefore, understanding the relationship between root architecture and above-ground growth, productivity, and N absorption may improve N efficiency and increase grain yield [12,13]. Root architecture is of great significance to above-ground growth and yield formation, especially nutrient absorption and use efficiency, and resistance to adversity [14]. While the root system and canopy compete for energy and nutrient resources [15], they also maintain the dynamic balance and development of plant growth [16,17]. Actually, substances absorbed by the root system and synthesized by the canopy are preferentially supplied to the growth of organs near the source under low N conditions [18]. It was urgent to shape an optimized population, and then balance and coordinate the metabolism of the root system and canopy. In addition, improving the root dry weight, root length, and its density may significantly enhance N uptake ability and grain yield [19,20]. Thus, it is essential to study root morphology characteristics and mechanisms of how N application rates regulate the root morphology in maize.

Root characteristics are essential to root growth, and their distribution significantly influences the uptake of nutrients and soil moisture [21], which directly affects crop growth and development [22]. Root morphological characteristics are closely related to root absorption, assimilation, and transport to above-ground. The root system includes root length, lateral root branching, root occurrence, three-dimensional distribution in space, root growth angle, root growth, and available soil water [23–25]. Anatomical structures such as root aeration tissue, cell layer number, and cell size affect root function and nutrient absorption [26–28]. Root aeration tissue could increase nutrient capture by reducing respiration and the nutrient content of root tissue, thus improving crops' adaptation to the nutrient environment [29]. N-efficiency maize varieties have much more underground nodes, longer length, and developed cortical ventilation tissue, which are beneficial to absorb more nitrogen from the soil [30–32]. N-efficiency maize varieties show a higher grain yield and dry matter accumulation capacity post-silking, as well as higher N use efficiency under a moderate N reduction level [33]. Thus, it is essential to study root architecture and anatomical structure characteristics in higher N-efficiency maize varieties under varying N application rates.

Root-bleeding sap is a sign of root pressure, plant growing potential, and root activity [34]. Root-bleeding sap is helpful to learn root behavior, especially nutrient and water uptake [35]. Phytohormones play a vital role in plant growth [31]. Auxin is the earliest discovered member of the plant hormone family, and it is very important for the maintenance of root stem cells [36,37]. Jasmonic acid plays an important role in plants resisting biotic and abiotic stresses and controls the formation of lateral roots by regulating biosynthesis and polar transport of auxin [38]. Thus, it is essential to study the changes of root-bleeding sap and endogenous auxin and jasmonic acid under varying N application rates. The mechanisms of how N application rates regulate root morphology and grain yield of spring maize by affecting root anatomical and endogenous hormone contents in roots remain unclear.

The purpose of this study was to address three questions: (1) How does the N application rate influence maize root morphology and anatomy? (2) How does the N application rate affect endogenous hormone levels in roots? (3) What is the relationship between root morphology, endogenous hormone levels, and grain yield under various N application rates?

## 2. Materials and Methods

### 2.1. Experimental Location

The field experiment was conducted in Changli, Hebei Province, China (39°07′ N, 119°17′ E) in 2019 and 2020. The soil texture for experiments is medium loam, with a bulk density of 1.44 g cm$^{-3}$, organic matter of 19.08 g kg$^{-1}$, total N of 1.68 g kg$^{-1}$, alkali-

hydrolysable N of 102.35 mg kg$^{-1}$, extractable Olsen-P of 23.59 mg kg$^{-1}$, and ammonium acetate extractable K of 74.10 mg kg$^{-1}$. Daily precipitation and mean temperature during the growing season are shown in Figure 1.

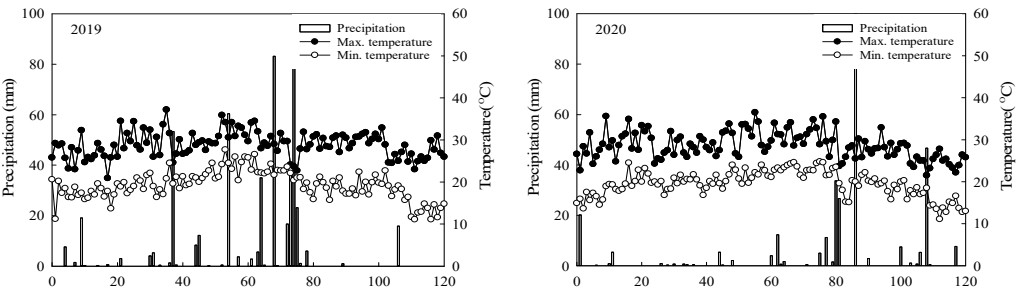

**Figure 1.** Daily precipitation and temperature during the growing seasons of 2019 and 2020.

## 2.2. Experimental Design

In this study, both field and pot experiments were designed. The pot experiment of $^{15}$N-labeled urea were carried out. The maize hybrid used in the study was Jingnongke 728, an N-efficiency maize variety.

### 2.2.1. Field Experiment

The field experiment adopted a randomized block design with three repetitions for each treatment in 2019 and 2020. The experiment includes three N rates: 0, 180, and 360 kg N ha$^{-1}$, referred to as zero-N (N0), optimized-N (N180), and higher-N (N360), respectively. The plot area measured was 24 m$^2$. Nitrogen, phosphorus, and potassium fertilizers were urea (containing N 46%), superphosphate (containing $P_2O_5$ 51%), sulfuric acid, and potassium (containing $K_2O$ 52%). All levels applied $P_2O_5$ 90 kg·ha$^{-1}$ and $K_2O$ 120 kg·ha$^{-1}$. The planting density was 75,000 plants·ha$^{-1}$. Other cultivation practices, such as irrigation (conducted prior to seeds being sown and at the heading stage), removal of weeds, and chemical control of diseases and pests, were performed using the recommended conventional approaches. The sowing dates were 30 and 29 May in 2019 and 2020, and the harvest dates were 20 and 21 September in 2019 and 2020, respectively.

### 2.2.2. Pot Experiment

There were 90 plastic pots with a height of 60 cm and a diameter of 32 cm in 2020. The source of the soil was from a field plot. There were three N rates: 0, 2.4, and 4.8 g N, for each pot (referred to as rates N0, N180, N360, respectively), and each N rate set 30 pots. Nitrogen, phosphorus, and potassium fertilizers were urea (containing N 46%), superphosphate (containing $P_2O_5$ 51%), sulfuric acid, and potassium (containing $K_2O$ 52%). All levels applied 1.2 g $P_2O_5$ and 1.6 g $K_2O$ for each pot. The $^{15}$N-labeled urea (abundance value 10.6%) application rate between the three N inputs was 48 mg plant$^{-1}$, and it was applied externally before sowing. The sowing and harvest dates were 30 May and 18 September in 2020, respectively.

## 2.3. Data Collection

### 2.3.1. Shoot Dry Matter Weight

In the field and pot experiments, whole above-ground plants were harvested at silking (R1) and maturity stages (R6). Each time, five plant samples were obtained from the center of each plot, and subsequently separated into the ear leaf, other leaves, stem (including sheath), cob, tassel, ear bracts, and grain. All samples were oven-dried at 105 °C for 30 min, and then dried at 70 °C to constant weight, thus obtaining the dry matter weight.

### 2.3.2. Root Morphology

In the field experiment, roots were sampled from the top down at 0–30 cm at silking (R1) and maturity stages (R6). Three plants randomly selected from the central three rows of each subplot were cut with a sickle, then the soil volume of 28 cm (intra-row spacing) in width $\times$ 60 cm (inter-row spacing) in length $\times$ 30 cm in depth surrounding each plant was excavated with a shovel [30].

In the pot experiment, roots were sampled from the top down in three levels, from 0–20, 20–40, to 40–60 cm. The soils were soaked in plastic pots and roots were stirred and poured into a sieve (0.25 mm$^2$ mesh). The sieve was suspended in a large water bath and shaken continuously until the roots were washed free of soil. Soil materials remaining on the sieve were removed manually. Roots from each core were then scanned using a scanner (EpsonV700, Beijing, China). These root images were subject to the software WinRHIZO version 5.0 (Regent Instruments Inc., Quebec, QC, Canada) for further analysis. The root length and root surface area were obtained. To obtain the constant root dry weight, the root samples was dried at 105 °C for 20 min, and then dried at 70 °C for 48 h.

### 2.3.3. Root-Bleeding Sap

Root-bleeding sap collection was carried out according to the methods by Yang [39], with melioration to better adapt to maize-bleeding sap collection. In the field and pot experiments, three plants were sampled at silking (R1), 30 days after silking (R3), and maturity stages (R6), where each plant was cut at an internode about 12 cm above the soil surface at 18:00 pm. A finger cot and rubber tube were placed over the plant stalk for draining the exudate into flasks. The finger cot was put on the upper end of the stalk and a rubber band was used to ensure that the finger cot was firmly fastened to the stalk. The other end of the rubber tube was plugged up with cotton. Each plant or sampling area was covered with a polyethylene sheet to keep dust and insects out. Bleeding sap in the flask was collected the next morning at 6:00 am and the volume was measured. The rate of delivery was calculated by multiplying the concentration by the volume of sap for each time. The delivery rate was expressed as concentration per time unit per root (g h$^{-1}$ root$^{-1}$) according to Wang et al. [40].

### 2.3.4. Root Microstructure and Ultrastructure

In the pot experiment, three samples were taken from the third nodal root at silking (R1) and 30 days after silking (R3), the root segments were cut 4–10 cm from the root base, and the length of the root tip was 1 cm. The samples were fixed with FAA fixation solution. Varying alcohol gradients were used for dehydration. The xylene gradient was transparent, and a biological tissue embedding machine (Jindi YD-6L, Jinhua, China) was used for embedding. Then, a paraffin microtome (Leica RM2255, Wetzlar, Germany) was used for sectioning. The slice thickness was 10 µm. The slices were double-stained with saffron and solid green and sealed with Canadian neutral gum for preservation. Then, the slices were observed and photographed with an optical microscope (Olympus SZX10, Tokyo, Japan). Three slices were observed in each replicate and the photography software captured images. Then, the Image-Pro Plus 6.0 software was used to analyze the root cortex aeration tissue (RCA), the thickness of cortical cells, the ratio of the cortex to root diameter, the central column diameter, and the duct diameter.

The samples were fixed in 2.5% glutaraldehyde (pH = 7.4) for 2 h. After washing three times with 0.1 M phosphate buffer (pH = 7.2), they were fixed in 1% osmic acid at 4 °C for 2 h. Then, the samples were gradient-dehydrated in graded series of ethanol. Subsequently, the samples were embedded in Epon-Aralditeresin for penetration and placed in a model for polymerization. After these samples were used for positioning, the ultra-thin samples were made and collected for microstructure analysis. Afterwards, a counter staining of 3% uranylacetate and 2.7% lead citrate was performed. Then, observation with a transmission electron microscope was completed (HITACHI HT7700, Tokyo, Japan).

### 2.3.5. Endogenous Hormones in Root

In the pot experiment, roots were sampled from three varying plants at the silking stage (R1), 30 days after silking (R3), and the maturity stage (R6), and the endogenous hormones in the root were measured. Abscisic acid (ABA) and jasmonic acid (JA) contents were determined by using ABA and JA assay kits (ABA-3-T and JA-3-T, Suzhou Comin Biotechnology Co., Ltd., Suzhou, China). Auxin (IAA) and gibberellins (GA) contents were determined by using GA, CTK, and ETH ELISA assay kits (Quanzhou Rui xin Biological Technology Co., LTD, Quanzhou, China) according to the manufacturer's instructions.

### 2.3.6. Total N Concentration and $^{15}$N Allocation

In the field and pot experiments, appropriate plant amounts (0.3 g) of ground powder were used to determine total N content by the modified Kjeldahl digestion method [41]. After that, nitrogen uptake efficiency was determined by the following formula [19,42]. Nitrogen uptake efficiency (NUPE) was calculated as Equation (1):

$$NUPE = \text{plant N content at maturity} / \text{N application rate} \qquad (1)$$

At the silking stage (R1) and the maturity stage (R6), we took the $^{15}$N-labeled plant samples, and then separated the roots, leaves, stems (stalks, sheaths, tassels), cobs, bracts, and grains. All samples were oven-dried at 105 °C for 30 min, and then dried at 70 °C to constant weight, and thus the dry matter weight was obtained. The sample was pulverized by a pulverizer and then ground with the MM400 high-efficiency biological sample preparation instrument (Retsch, Arzberg, Germany). The $^{15}$N abundance of the plant organs was measured with wisoprime100 mass spectrometer (Isoprime, Manchester, UK).

### 2.3.7. Grain Yield and Its Components

In the field experiment, twenty ears were collected from the center rows of each plot at the maturity stage (R6). They were used to investigate grain yield and its components. Kernel numbers per plant were counted for all harvested ears. Then, 3 samples of 1000 kernels were oven-dried at 70 °C for 72 h to constant weight and weighed to estimate 1000-weights. Grain yield was expressed at 14% moisture.

In the pot experiment, three ears were collected from each plot at the maturity stage (R6). They were used to investigate the single-plant yield.

### *2.4. Statistical Analysis*

The statistical differences of grain yield and root morphology (length, surface area, etc.) were tested using the least significant difference (LSD) of one-way ANOVA (SPSS 19.0, IBM Co., New York, NY, USA) at the $p < 0.05$ probability level. All figures were constructed using SigmaPlot 10 (Systat Software Inc., San Jose, CA, USA). The images and data of root anatomical slices were analyzed by Pro Plus 6.0.

## 3. Results

### *3.1. Root Morphology Characteristics*

N rate affected the root morphology of spring maize in 2020 (Figure 2). The largest difference of total root length and surface area between the three N application rates was seen at the R1 stage. Compared with N0, total root length and surface area increased by 51.36% and 42.58% under N180, and increased by 7.80% and 30.14% under N360, respectively ($p \leq 0.05$). In the soil layer of 0–20 cm, the proportions of root length in N0, N180, and N360 were 31.89%, 41.32%, and 32.87%, respectively. In the soil layer of 20–60 cm, the proportions of root length in N0, N180, and N360 were 68.11%, 67.13%, and 58.68%, respectively.

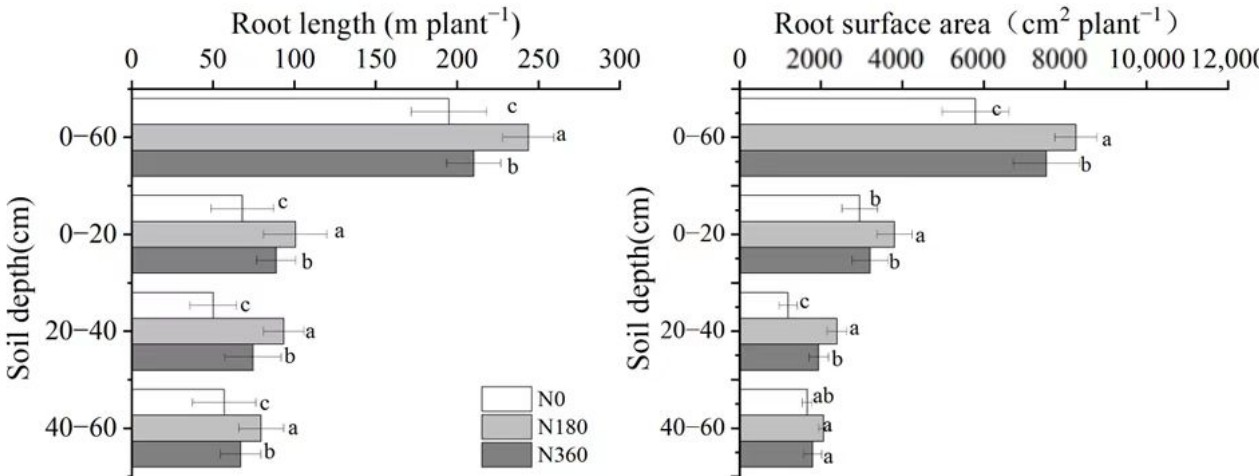

**Figure 2.** Effect of N application rates on the distribution of root length, root surface area, root volume, and root dry weights of maize at the silking stage (R1) in the pot experiment condition in 2020. N0, N180, and N360 indicate 0, 180, and 360 kg N ha$^{-1}$, respectively. N treatments followed by different letters indicate significant differences at $p < 0.05$ according to LSD.

### 3.2. Root Microstructure and Ultrastructure Characteristics

N rate affected the root microstructure of spring maize at R1 and R3 stages in 2020 (Table 1 and Figure 3). At the R1 stage, the RCA in N180 and N360 decreased by 32.42% ($p > 0.05$) and 11.92% ($p \leq 0.05$) compared with N0, while the root diameter and duct diameter increased (Table 1). At the R3 stage, the RCA in N180 and N360 decreased by 3.80% ($p \leq 0.05$) and 21.49% ($p > 0.05$) compared with N0, while the diameter of the duct increased (Table 1).

**Table 1.** Root microstructure of maize at silking (R1) and 30 days after silking (R3) by different N application rates in the pot experiment condition in 2020.

| Growth Stage | N Rate | Thickness of the Cortex (μm) | Ratio of Root Cortex Aeration Tissue (RCA, %) | Central Column Diameter (μm) | Ratio of the Cortex to the Root Diameter (%) | Vessel Diameter (mm) |
|---|---|---|---|---|---|---|
| R1 | N0 | 277.87b | 14.00a | 326.40c | 41.86a | 67.97c |
|  | N180 | 247.79c | 12.33b | 356.04b | 41.99a | 83.46b |
|  | N360 | 341.19a | 9.46c | 466.30a | 41.03a | 91.68a |
| R6 | N0 | 208.20c | 13.68a | 295.32a | 38.26a | 48.55b |
|  | N180 | 341.47a | 13.16a | 290.18a | 38.71a | 43.38b |
|  | N360 | 290.68b | 10.74b | 233.54b | 27.54b | 57.20a |

Different letters indicate significant differences between treatments with N application rates ($p < 0.05$).

N rate affected the root tip ultrastructure of spring maize at the R1 stage (Figure 4). Compared with N0, the root tip cell structure was intact in the N180, and there was no separation of the cytoplasm and cell wall, and the organelles were intact, without swelling and vacuolization. The cell membrane of some cells was broken, and the degradation disappeared. The root tip cell wall structure was complete, and the cell wall was separated in the N360, and the organelles were intact, without swelling and vacuolization.

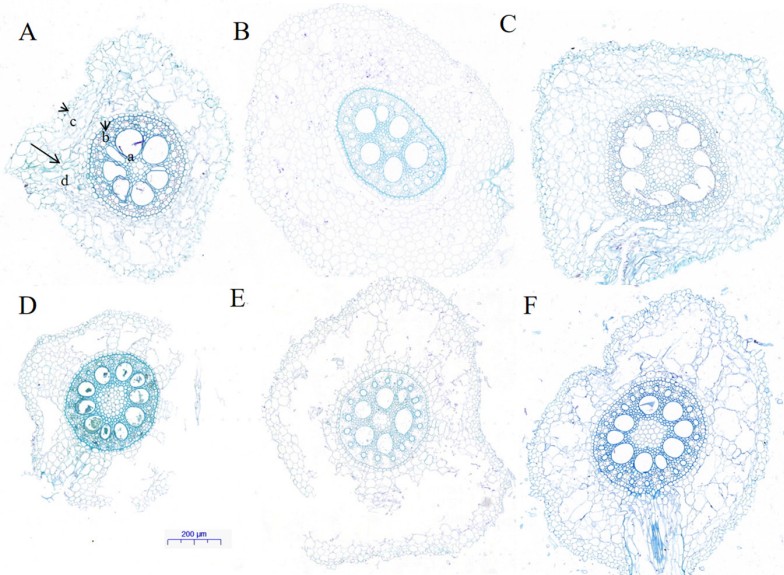

**Figure 3.** Effects of N application rates on root microstructure of maize in the pot experiment condition in 2020. (**A–C**) N0, N180, and N360, respectively, at the silking stage (R1). (**D–F**) N0, N180, and N360, respectively, at 30 days after silking (R3). a: Stele, b: vessel, c: cortex, and d: root cortex aeration tissue.

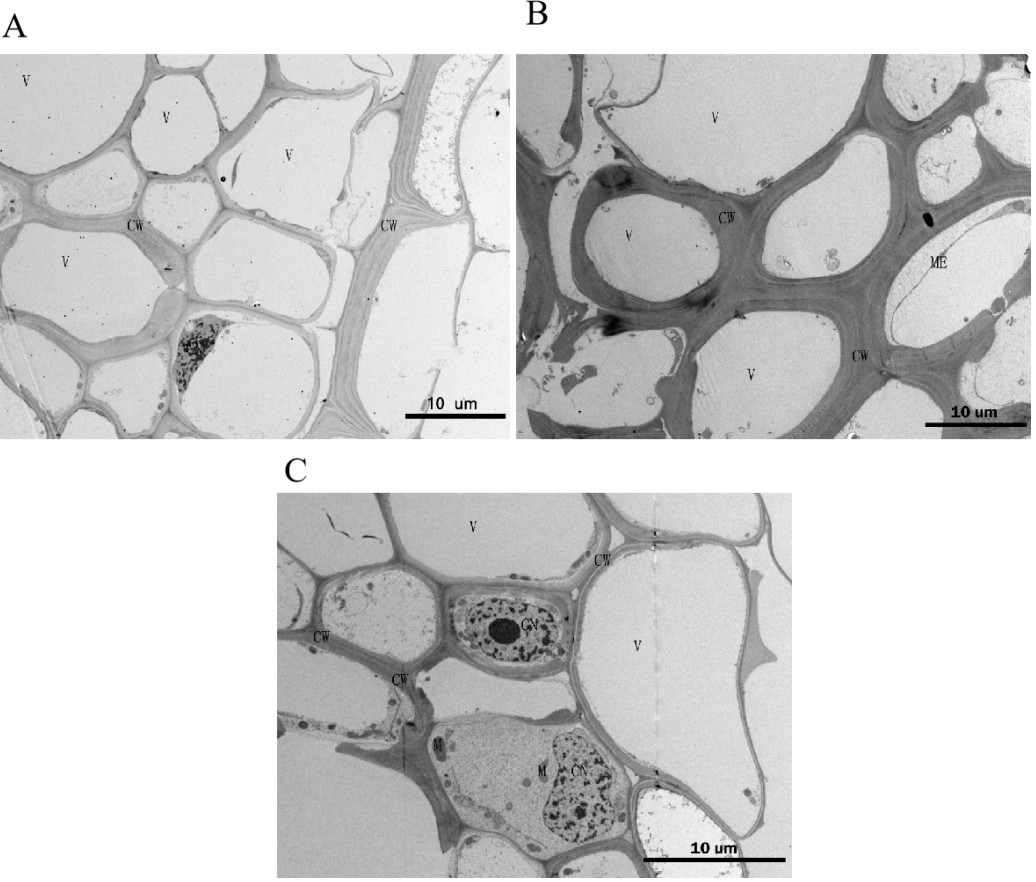

**Figure 4.** Effects of N application rates on root tip cell ultrastructure of maize in the pot experiment condition in 2020. (**A–C**) N0, N180, and N360, respectively, at the silking stage (R1). C, cytoplasm; M, mitochondrion; ME, cell membrane; CN, cell nucleus; CW, cell wall; P, plastid; PC, parenchyma cell; V, vacuole.

### 3.3. Root-Bleeding Sap

N application rates affected the root-bleeding sap of spring maize in 2020 (Figure 5). In the field experiment condition, the root-bleeding sap in N180 and N360 increased by 21.91% ($p \leq 0.05$) and 39.41% ($p \leq 0.05$) compared with N0 at the R1 stage, and the root-bleeding sap in N180 and N360 increased by 41.25% ($p \leq 0.05$) and 82.84% ($p \leq 0.05$) compared with N0 at the R6 stage. The root-bleeding sap change was similar in pot and field experiment conditions. The root-bleeding sap decreased at the R3 stage, and those values in N0, N180, and N360 decreased by 11.46% ($p \leq 0.05$), 12.27% ($p \leq 0.05$), and 36.42% ($p \leq 0.05$), respectively.

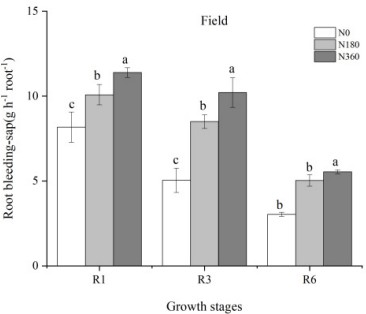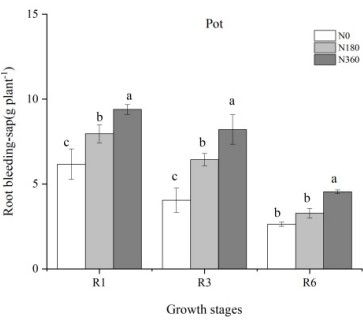

**Figure 5.** Effects of N application rates on root-bleeding sap in field and pot experiment conditions in 2020. N0, N180, and N360 indicate 0, 180, and 360 kg N ha$^{-1}$, respectively. N treatments followed by different letters indicate significant differences at *p* < 0.05 according to LSD.

### 3.4. Endogenous Hormone Levels in Root

N application rates affected the endogenous hormone levels in roots in 2020 (Figure 6). At the R1 stage, the auxin, gibberellins, abscisic acid, and jasmonic acid levels in the root increased with the increase from the N180 rate up to N360 ($p \leq 0.05$). The average values of endogenous auxin, gibberellins, abscisic acid, and jasmonic acid levels under N180 and N360 increased by 41.95%, 38.69%, 25.15%, and 33.51% compared with N0. Compared with the R1 stage, the gibberellins and jasmonic acid levels in the root decreased at R3 and R6 stages, whereas the auxin and abscisic acid levels in the root increased.

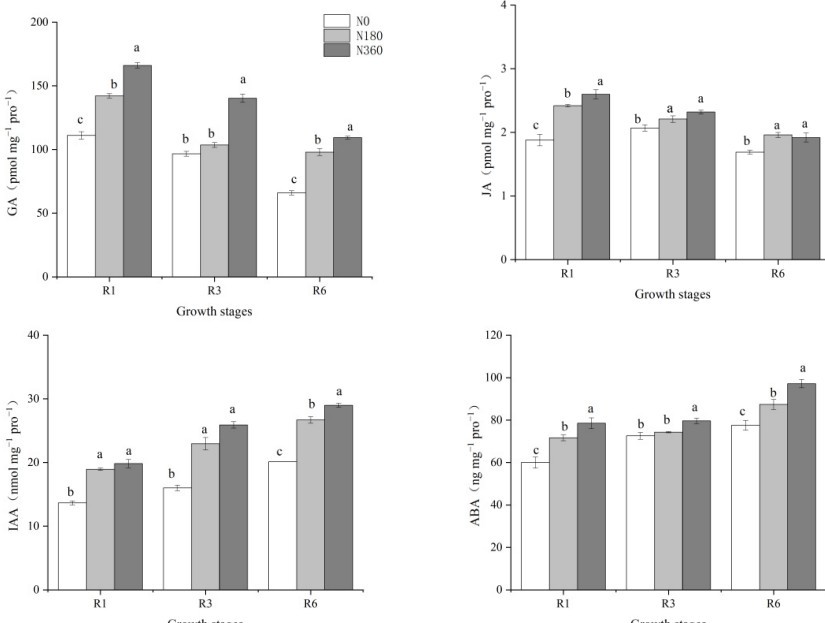

**Figure 6.** Effects of N application rates on the endogenous hormone contents of maize root in the pot experiment condition in 2020. N0, N180, and N360 indicate 0, 180, and 360 kg N ha$^{-1}$, respectively. N treatments followed by different letters indicate significant differences at *p* < 0.05 according to LSD.

### 3.5. Total N Concentration and $^{15}$N Allocation

N application rates affected the plant N content of spring maize in 2020 (Table 2). In the field experiment condition, the plant total N content of N180 and N360 increased by 2.40% ($p > 0.05$) and 14.46% ($p \leq 0.05$) compared with N0 at the R1 stage. Plant total N content of N180 and N360 were 8.88% and 36.77%, significantly higher than that of N0 at the R6 stage. In the pot experiment condition, the plant total N content of N180 and N360 increased by 7.1% and 23.87% ($p \leq 0.05$) compared with N0 at the R1 stage, and the plant N content of N180 and N360 were 4.05% ($p > 0.05$) and 36.33% ($p \leq 0.05$) higher than that of N0 at the R6 stage. Post-silking, the plant N and grain N contents increased with the increase from N180 up to N360, and N uptake efficiency decreased from N180 to N360. Nitrogen uptake efficiency in N180 was 65.33% and 57.89% higher than N360 in field and pot experiment conditions, respectively.

**Table 2.** Total N uptake at silking (R1) and maturity stages (R6), total N uptake of maize from R1 to R6, grain N content, and nitrogen uptake efficiency by different N application rates in 2020.

| Experiment Condition | N Rate | R1 N Uptake (g Plant$^{-1}$) | R6 N Uptake (g Plant$^{-1}$) | Post-Silking N Uptake (g Plant$^{-1}$) | Grain N Content (g Plant$^{-1}$) | Nitrogen Uptake Efficiency (kg kg$^{-1}$) |
|---|---|---|---|---|---|---|
| | N0 | 1.66b | 2.91b | 1.25c | 1.90c | - |
| Field | N180 | 1.70b | 3.19b | 1.49b | 2.23b | 1.24a |
| | N360 | 1.90a | 3.98a | 2.00a | 3.06a | 0.75b |
| | N0 | 1.69c | 3.46b | 1.77b | 2.42b | - |
| pot | N180 | 1.81b | 3.61b | 1.80b | 2.43b | 1.50a |
| | N360 | 2.22a | 4.55a | 2.17a | 2.86a | 0.95b |

Different letters indicate significant differences between treatments with N application rates ($p < 0.05$).

According to the $^{15}$N isotope labeling experiment (Table 3), the distribution proportion of $^{15}$N in root, ear leaves, other leaves, and the stem increased with the increasing N application rate at the R6 stage, whereas the distribution proportion of $^{15}$N in grain decreased. The distribution proportion of $^{15}$N in root of N180 and N360 increased by 35.44% ($p \leq 0.05$) and 23.09% ($p \leq 0.05$) compared with N0, and the distribution proportion of $^{15}$N in grain of N180 and N360 decreased by 10.97% ($p \leq 0.05$) and 16.11% ($p \leq 0.05$), respectively.

**Table 3.** $^{15}$N distribution in different organs of maize at silking (R1) and maturity stages (R6) by different N application rates in the pot experiment condition in 2020.

| Stages | N Rate | $^{15}$N Distribution in Different Organs (%) | | | | | | |
|---|---|---|---|---|---|---|---|---|
| | | Root | Ear Leaf | Other Leaves | Stem | Cob | Ear Bracts | Grain |
| | N0 | 8.21a | 7.45a | 41.32a | 43.02b | - | - | - |
| R1 | N180 | 7.09a | 6.84b | 36.23b | 57.73a | - | - | - |
| | N360 | 4.36b | 6.45b | 34.18b | 59.38a | - | - | - |
| | N0 | 1.21a | 0.96b | 7.61b | 10.02b | 0.21c | 0.38c | 79.61a |
| R6 | N180 | 0.93b | 0.95b | 7.47b | 9.09b | 0.47b | 0.43b | 81.60a |
| | N360 | 0.91b | 1.34a | 11.33a | 13.35a | 0.83a | 0.99a | 72.16b |

Different letters indicate significant differences between treatments with N application rates ($p < 0.05$).

### 3.6. Root/Shoot and Grain Yield

N rate affected the root/shoot of spring maize in 2019 and 2020 (Table 4). In the field and pot experiment conditions, root dry weight and root/shoot in N180 and N360 significantly increased compared with N0, and root dry weight in the soil layer of 20–40 cm was highest in N360, while there was no significant difference in soil layers of 0–20 and 40–60 cm between N180 and N360. N rate affected the grain yield of spring maize (Table 5). In the field experiment condition, average grain yield over two years in N180 and N360 was 20.44% and 16.61% higher than that of N0, respectively. The average kernel numbers

over two years in N180 and N360 were also higher than that of N0. In the pot experiment condition, average grain yield in N180 and N360 was 23.27% and 16.12% higher than that of N0, respectively.

**Table 4.** Shoot and root biomass and root/shoot of maize at silking (R1) and maturity (R6) stages under different N application rates.

| Years | Stages | N Rate | Field | | | Pot | | |
|---|---|---|---|---|---|---|---|---|
| | | | Shoot Biomass (g Plant$^{-1}$) | Root Biomass (g Plant$^{-1}$) | Root/Shoot | Shoot Biomass (g Plant$^{-1}$) | Root Biomass (g Plant$^{-1}$) | Root/Shoot |
| 2019 | R1 | N0 | 105.59b | 19.53b | 0.18b | - | - | - |
| | | N180 | 126.92a | 32.62a | 0.26a | - | - | - |
| | | N360 | 125.57a | 31.77a | 0.25a | - | - | - |
| | R6 | N0 | 345.97b | 18.02b | 0.05a | - | - | - |
| | | N180 | 362.76b | 25.94a | 0.07a | - | - | - |
| | | N360 | 381.58a | 24.70a | 0.06a | - | - | - |
| 2020 | R1 | N0 | 113.99b | 21.53c | 0.19b | 122.71c | 30.60b | 0.29ab |
| | | N180 | 131.86a | 35.62ab | 0.27a | 135.61b | 40.69a | 0.36a |
| | | N360 | 131.72a | 34.77b | 0.26a | 149.09a | 39.30a | 0.31a |
| | R6 | N0 | 266.44c | 20.02b | 0.07a | 290.76c | 20.50c | 0.07b |
| | | N180 | 303.67b | 27.94a | 0.09a | 317.10b | 31.50a | 0.10a |
| | | N360 | 311.02a | 26.70a | 0.08a | 355.24a | 25.50b | 0.07b |

Different letters indicate significant differences between treatments with N application rates ($p < 0.05$).

**Table 5.** Effects of N application rates on grain yield, kernel numbers, and 1000-weight of maize in field and pot experiment conditions.

| Years | N Rate | Field | | | | Pot | | |
|---|---|---|---|---|---|---|---|---|
| | | Ears Numbers (Ear ha$^{-1}$) | Kernel Numbers (No.) | 1000-weight 1001-(g) | Grain Yield (kg ha$^{-1}$) | Kernel Numbers (No.) | 1000-Weight (g) | Grain Yield (g Plant$^{-1}$) |
| 2019 | N0 | 59,944.4b | 482.6c | 310.2b | 8889.40b | - | - | - |
| | N180 | 68,523.5a | 506.4b | 329.8a | 10,319.18a | - | - | - |
| | N360 | 68,982.4a | 524.2a | 326.3a | 9899.97a | - | - | - |
| 2020 | N0 | 56,944.4b | 486.1b | 310.2b | 9253.2b | 466.7b | 318.2a | 169.56b |
| | N180 | 68,518.5a | 517.1a | 326.4a | 11,531.3a | 496.4a | 320.2a | 209.01a |
| | N360 | 68,381.5a | 509.8a | 322.8a | 11,255.4a | 508.6a | 332.6a | 196.89a |

Different letters indicate significant differences between treatments with N application rates ($p < 0.05$).

*3.7. Correlation Analysis*

To determine whether various factors were correlated (Figure 7), the correlations between root phytohormones, anatomical structure, and root morphology traits and NUPE were examined. A significant positive correlation was found between IAA and RSD, RS, as well as RV, and then a positive correlation were found between RS, RV, and grain yield, as well as NUPE, but a negative correlation was found between RCA and grain yield, as well as NUPE.

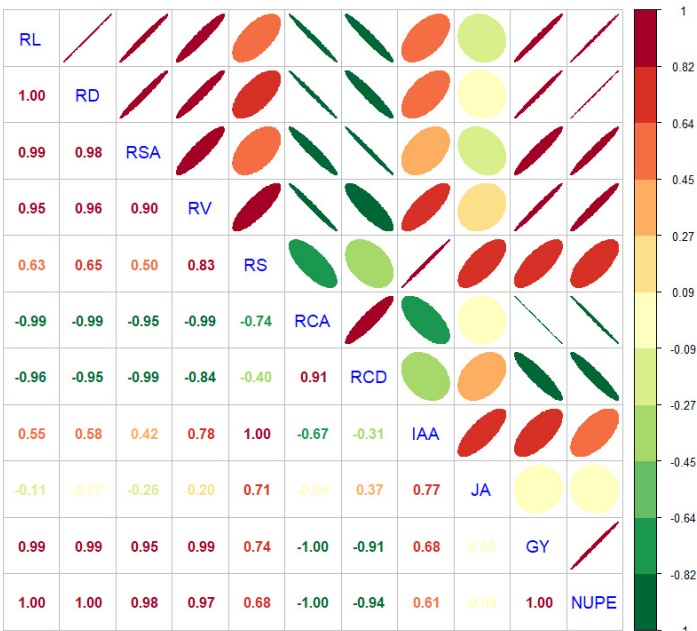

**Figure 7.** Correlations combined with a significant test between root morphology, root dry weights, endogenous hormone contents, and N uptake efficiency and grain yield. The darker the color in the legend, the greater the correlation coefficient. Note: RL, root length; RD, root density; RV, root volume; RS, root surface area; RCA, ratio of root cortex aeration tissue; RCD, ratio of the cortex to the root diameter; IAA, auxin; JA, jasmonic acid; GY, grain yield; NUPE, N uptake efficiency.

## 4. Discussion

In maize production, higher grain yield is generally achieved through increased N fertilizer [43,44]. Here, the grain yield, root/shoot, and root dry weight under N360 and N180 conditions were significantly higher than those under N0 treatment in the spring maize hybrid tested. Previous studies demonstrated that grain yield could be altered by root morphology and root/shoot [12,33]. Root morphology traits are important supports for N uptake and thus affect transporting assimilates produced from source to grain.

N fertilizer plays an important role in regulating root growth [43]. The quest for improving N uptake efficiency has led to numerous studies on the physiology of how maize roots access, transport, and then utilize accumulated nitrogen [45–47]. It has been reported that appropriate reduction of the nitrogen (N) fertilizer rate can increase crop yield and reduce greenhouse gas emissions [48–51]. Nutrient availability in the soil influenced both root growth and the pattern of root development [14,30,52]. In this study, the total root length and surface area increased by 51.36% and 42.58% under the optimized-N condition compared with N0 and increased by 7.80% and 30.14% under the higher-N condition compared with N0, respectively. According to the [15]N isotope labeling experiment, the distribution proportion of [15]N in the root of N180 and N360 increased by 35.44% and 23.09% compared with N0, while the distribution proportion of [15]N in grain of N180 and N360 decreased by 10.97% and 16.11%, respectively. Our results suggested that the spring maize variety had a longer total root length and better root distribution in optimized-N conditions than other treatments, indicating that optimized-N application could attain shoot growth potential and thus formed a larger root system.

Additionally, root characteristics were close to the root anatomical structure. Root anatomical structure can affect the root morphology, the root aeration tissue, the cell layer number, and the cell size, and other anatomical structures affected root function and nutrient absorption [26–28]. The role of root aeration tissue could increase nutrient capture by reducing the nutrient content of respiration and improve root growth under low phosphorus availability in maize [52]. In this study, the RCA in optimized-N and higher-N conditions decreased by 32.42% and 11.92% compared with N0 and optimized-N

increased the RCA formation compared with the higher-N condition. Our study results were consistent with those of a previous study, which reported that RCA could improve plant growth under N-limiting conditions by decreasing root metabolic costs [29]. We found that post-silking plant total N and grain N contents under N360 were higher than that of N180, whereas N uptake efficiency decreased from N180 to N360. Nitrogen uptake efficiency in N180 was 65.33% and 57.89% higher than that of N360 in field and pot experiment conditions, respectively, and this is the reason that RCA could enhance soil exploration and N acquisition in deep soil strata [28]. Therefore, optimized-N could improve the ratio of root cortex aeration tissue (RCA) of spring maize, indicating that increased RCA formation appears to be a promising breeding target for enhancing crop N acquisition.

Moreover, the phytohormones played an important role in coordinating lateral roots' growth and geotropism of plants [31,53]. Auxin (IAA) was very important for the maintenance of root stem cells, which was mainly achieved by inducing the accumulation of transcription factor PLT1/2 [36,37]. Additionally, jasmonic acid was an important lipid hormone in plants. Jasmonic acid can inhibit taproot elongate and induce the growth of lateral roots, as well as control the formation of lateral roots by regulating the biosynthesis and polar transport of auxin [38]. In this study, N rate affected the endogenous hormone levels in the root, the auxin and jasmonic acid levels in the root increased from N180 to N360. A significant positive correlation was found between auxin and root length, root surface area, as well as root volume. We also found a positive correlation between RCA and NUPE, indicating that optimized-N application decreased root cortical burden and root metabolic costs, and improved root length and N uptake ability in controlled environments, as well as emphasized the significance of root cortex aeration tissue (RCA) of maize. Overall, the knowledge provides pivotal physiology evidence for elevating root morphogenesis of the spring maize variety, which is helpful to synergistically improve maize yield and nitrogen use efficiency.

## 5. Conclusions

Improved maize grain yield relied heavily on root morphogenesis, which was determined by plant type. The largest difference in total root length and surface area among the three N application rates was found at the silking stage. N application affected the ultrastructure and the tip cell structure of the root. Optimized-N decreased root cortical burden and root metabolic costs and improved the root length and root surface area through regulating auxin and jasmonic acid levels and affected N uptake and grain yield of an N-efficient spring maize variety. Our study is helpful for understanding the characterization of root morphology and the physiological structure of spring maize varieties and was designed to help satisfy the intense demand for improved crop production.

**Author Contributions:** X.L. collected the samples, analyzed the samples, and wrote the manuscript; R.W., J.W., B.Z., X.W. and M.Z. contributed to writing and editing of the manuscript; C.L. contributed to the design of the work, analysis, and revised the manuscript. All authors have read and agreed to the published version of the manuscript.

**Funding:** This study was supported by the National Natural Science Foundation of China (No. 31971852, 2019), the China Agriculture Research System of MOF and MARA (CARS-02-14), and the Scientific Research Fund of Hebei Normal University of Science and Technology (2020YB005).

**Data Availability Statement:** Not applicable.

**Conflicts of Interest:** The authors declare no conflict of interest.

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
