# Peer review of "Characterization of Root Morphology and Anatomical Structure of Spring Maize under Varying N Application Rates and Their Effects on Yield"

_agronomy, doi:10.3390/agronomy12112671_

Round 1

Reviewer 1 Report

I think that this study was designed appropriately and conducted according to high scientific and technical requirements.

In my opinion, this article is suitable for publication in Agronomy, and it can be accepted for publication after minor revisions, mostly technical. My suggestions for correction are as follows:

In MM, in the section Field experiment, the development phase in which the measurements were made should be indicated (R1, R3, R6), or there are more of them, considering that the R5 stage in line 154 is mentioned, or it is a mistake and it is R6 stage. Please correct.

Line 107: Filed experiment… should Field experiment

In the text, joined words appear several times. I will indicate the lines where I noticed the connected words, but please check the whole text carefully!

Line 154: silking(R1)

Line 229: maizeat

Line 233: tissue(RCA)

262 and 283: …, At R1, - at R1

267: 12.27%(….

280: Table3

281: (P>0.05)and 

290: Table3

293: 35.44%(P…

297: ratesatbucket

361 and 378: Table1

376 and 377: 15N – you put 15 in superscript, but not in MM and results, so please show the same way.

422: ….insecurity.Disaster…..

430: ….from1970...

444: ….Agriculture,2019

482: …Plant physiology,2014…..

500: Da,2014; 2014 – bold

525: …hybrids.Plant Soil , - Please correct!

References:

422: del – Del

423: 2021 – bold

426: Journal title – should be in italic; 2011 - bold

444: The title of the article is missing. Please add!

539: 2015 - bold

Author Response

  1. The reviewer’s comment:

In MM, in the section Field experiment, the development phase in which the measurements were made should be indicated (R1, R3, R6), or there are more of them, considering that the R5 stage in line 154 is mentioned, or it is a mistake and it is R6 stage. Please correct.

The authors’ Answer:

In MM, R1 indicate the silking stage of maize, R3 indicate 30 days after silking stage of maize, R6 indicate the maturity stage of maize.

  1. 2. The reviewer’s comment:

Line 107: Filed experiment… should Field experiment

The authors’ Answer:

We had revised Field experiment.

  1. 3. The reviewer’s comment:

In the text, joined words appear several times. I will indicate the lines where I noticed the connected words, but please check the whole text carefully!

Line 154: silking(R1)

Line 229: maizeat

Line 233: tissue(RCA)

262 and 283: …, At R1, - at R1

267: 12.27%(….

280: Table3

281: (P>0.05)and 

290: Table3

293: 35.44%(P…

297: ratesatbucket

361 and 378: Table1

376 and 377: 15N – you put 15 in superscript, but not in MM and results, so please show the same way.

422: ….insecurity.Disaster…..

430: ….from1970...

444: ….Agriculture,2019

482: …Plant physiology,2014…..

500: Da,2014; 2014 – bold

525: …hybrids.Plant Soil , - Please correct!

 References:

422: del – Del

423: 2021 – bold

426: Journal title – should be in italic; 2011 - bold

444: The title of the article is missing. Please add!

539: 2015 - bold

The authors’ Answer:

The mistake of above in the MS, we had revised. We also carefully cheked other mistake of the MS.  

Reviewer 2 Report

Dear authors, I have finalized the analysis of the manuscript entitled "Characterization of root morphology, anatomical structure of spring maize under varying N application rates and their effects on yield". There are some comments and suggestions that I consider will improve your work.

Please use an English Professional editing service to analyze the text of the manuscript. there are parts of the text that are hard to read and understand. E.g Line 57-62. - structures like canopy population, It was urgent to shape an optimized population, improve root dry weight....may significantly enhance N uptake. The last one you can rewrite like - An improved N uptake ability and grain yield was observed in plants with higher root length and density.

Line 81-82 - Root bleeding gap or sap. Please correct in the text

Mat and Meth section

You need to add a paragraph at the beginning of this section, where to explain why do you have two types of experiments - field and pot - please change bucket to pot - it is interesting to have an experiment replication in controlled conditions, but you need to clearly state this and the reason. Also, you need to synchronize the sample collection in both type of experiments. In each of the sub-sections of both experiments you need to say why did you do a number of measurements/analyzes. 

An idea for mat and math section - present all analysis in one table and simply add a X in which experiment you performed.

Line 107 - Change filed to field

You do:

2.3.3. Root bleeding sap in R1, R3 and R5 (line 154)

2.3.4 Root microstructure and ultrastructure in R1 and R3 (line 166)

2.3.5 Endogenous hormones in root in R1, R3 and R6 (line 187)

2.3.6 Total N concentration and 15N allocation in R1 and R6 (line 199)

Explain the differences and missing.

3 Results

3.1 Root morphology characteristics

line 223 - you say "in 2019 and 2020, and increased 7.80% and 30.14% under" from figure 2. But the Figure 2 caption is "Figure 2. Effect of N rates on the distribution of root length, root surface area, root volume and root dry weights of maizeat silking stage (R1) in bucket condition in 2020." - explain the difference and correct in the text

3.2 Root microstructure and Ultrastructure Characteristics 

This sub-section last for 14 lines and present 2 tables and 2 Figures - expand it to an appropriate length. This will help you to construct the discussion section based on the explanation of your observations and results.

The same observation - expand the interpretation of your tables and figures - is necessary along the entire results section. You have a lot of data and they deserve to be presented.

Discussion section

Move all the text that is referenced to a previous table or figure in the results section. Do not add figures in this section, just focus on comparing and discuss your own results related to international literature.

Conclusion should be rewritten to present your main findings, with values and numbers.

Another suggestion is to make a clear separation between field and pot conditions. This will present better your results. 

I like your idea, but it needs to be improved in order to present well your research. 

Author Response

  1. The reviewer’s comment:

Please use an English Professional editing service to analyze the text of the manuscript. there are parts of the text that are hard to read and understand. E.g Line 57-62. - structures like canopy population, It was urgent to shape an optimized population, improve root dry weight....may significantly enhance N uptake. The last one you can rewrite like - An improved N uptake ability and grain yield was observed in plants with higher root length and density.

The authors’ Answer:

We had carefully polish the languages of the MS.  

  1. 2. The reviewer’s comment:

Line 81-82 - Root bleeding gap or sap. Please correct in the text

The authors’ Answer:

We had correct the root bleeding sap in the text

  1. 3. The reviewer’s comment:

Mat and Meth section

You need to add a paragraph at the beginning of this section, where to explain why do you have two types of experiments - field and pot - please change bucket to pot - it is interesting to have an experiment replication in controlled conditions, but you need to clearly state this and the reason. Also, you need to synchronize the sample collection in both type of experiments. In each of the sub-sections of both experiments you need to say why did you do a number of measurements/analyzes.

The authors’ Answer:

In this study, both field and pot experiment were design. the pot experiment was in controlled conditions, and can design the 15N-labeled urea. The maize hybrid used in the study was Jingnongke 728, a N-efficiency varieties.

  1. 4. The reviewer’s comment:

An idea for mat and math section - present all analysis in one table and simply add a X in which experiment you performed.

The authors’ Answer:

The idea is very good, we had try to present all analysis in one table, but failed.

  1. 5. The reviewer’s comment:

Line 107 - Change filed to field

You do:

2.3.3 Root bleeding sap in R1, R3 and R5 (line 154)

2.3.4 Root microstructure and ultrastructure in R1 and R3 (line 166)

2.3.5 Endogenous hormones in root in R1, R3 and R6 (line 187)

2.3.6 Total N concentration and 15N allocation in R1 and R6 (line 199)

Explain the differences and missing.

The authors’ Answer:

We change filed to field in the text.

2.3.3. Root bleeding sap in R1, R3 and R5 should change of root bleeding sap in R1, R3 and R6.

2.3.4 Root microstructure and ultrastructure in R1 and R3

2.3.5 Endogenous hormones in root in R1, R3 and R6

2.3.6 Total N concentration and 15N allocation in R1 and R6

  1. 6. The reviewer’s comment:

line 223 - you say "in 2019 and 2020, and increased 7.80% and 30.14% under" from figure 2. But the Figure 2 caption is "Figure 2. Effect of N rates on the distribution of root length, root surface area, root volume and root dry weights of maizeat silking stage (R1) in bucket condition in 2020." - explain the difference and correct in the text

The authors’ Answer:

We had correct the mistake in the text.

N rate affected root morphology of spring maize in 2020(Figure 2). Largest difference of total root length and surface area between three N application rates showed at R1 stage. Compared with N0, total root length and surface area increased 51.36% and 42.58% under N180, and increased 7.80% and 30.14% under N360, respectively (P≤0.05).

Figure 2. Effect of N application rates on the distribution of root length, root surface area, root volume and root dry weights of maize at silking stage (R1) in pot experiment condition in 2020.

  1. 7. The reviewer’s comment:

This sub-section last for 14 lines and present 2 tables and 2 Figures - expand it to an appropriate length. This will help you to construct the discussion section based on the explanation of your observations and results.

The same observation - expand the interpretation of your tables and figures - is necessary along the entire results section. You have a lot of data and they deserve to be presented.

The authors’ Answer:

We had correct in the text, condense it to an appropriate length.This sub-section last for 12 lines and present 1 tables and 2 Figures. We only presente the main text in this section.

  1. 8. The reviewer’s comment:

Move all the text that is referenced to a previous table or figure in the results section. Do not add figures in this section, just focus on comparing and discuss your own results related to international literature.

The authors’ Answer:

We move all the text that is referenced to a previous table or figure in the results section.

  1. 10. The reviewer’s comment:

Conclusion should be rewritten to present your main findings, with values and numbers.

The authors’ Answer:

The main findings had added in the conclusion.

  1. 11. The reviewer’s comment:

Another suggestion is to make a clear separation between field and pot conditions. This will present better your results.

The authors’ Answer:

In this study, both field and pot experiment were design. The pot experiment was in controlled conditions, and can design the 15N-labeled urea. In the resluts, we make a clear separation between field and pot conditions.

Reviewer 3 Report

The article presents the effect of nitrogen fertilization on root morphology. An important topic but widely reported in the available literature. 

The authors should indicate innovation: what makes their research different from those already known.

The authors present a lot of results, but describe them too poorly. E.g. Figure 1 how it influenced the research presented. The results of the field and bucket experiment are presented but not compared at all, why these experiments were conducted simultaneously. 

To be supplemented is the discussion: no scientific overtones, no literature review.

Author Response

  1. 1. The reviewer’s comment:

The article presents the effect of nitrogen fertilization on root morphology. An important topic but widely reported in the available literature. The authors should indicate innovation: what makes their research different from those already known.

The authors’ Answer:

There are three point innovation in this paper

  1. Largest difference in total root length and surface area among three N rateswas found at silking stage.
  2. N application rates also affected root microstructure and ultrastructure.The proportions of root aerating tissue under N180 and N360 were decreased at silking.
  3. Optimized-N application decreased root aerated tissue and improved root length and root surface area through regulating auxin and jasmonic acid levels, and affects N uptake and grain yield of an N-efficient spring maize variety.
  4. 2. The reviewer’s comment:

The authors present a lot of results, but describe them too poorly. E.g. Figure 1 how it influenced the research presented. The results of the field and bucket experiment are presented but not compared at all, why these experiments were conducted simultaneously. 

The authors’ Answer:

In this study, both field and pot experiment were design. the pot experiment was in controlled conditions, and can design the 15N-labeled urea. Also, we had correct in the part of result.

  1. 3. The reviewer’s comment:

To be supplemented is the discussion: no scientific overtones, no literature review.

The authors’ Answer:

In the discussion, we had added the scientific overtones and literature review.

Round 2

Reviewer 2 Report

Dear authors, the manuscript present multiple improvements.

Author Response

We are very grateful to reviewers for their very constructive and helpful suggestions.

Reviewer 3 Report

Accept.

Author Response

(The authors gave the same response as above.)
